# Laparoscopic Abdominal Surgery after Primary Breast Reconstruction Using an Abdominal Flap

**DOI:** 10.3390/medicina57090952

**Published:** 2021-09-10

**Authors:** Tsuyoshi Nakagawa, Goshi Oda, Hiroki Mori, Noriko Uemura, Kimio Wakana, Noriko Oshima, Masanori Tokunaga, Yuya Sato, Kumiko Hayashi, Yuichi Kumaki, Toshiaki Ishikawa, Kentaro Okamoto, Hiroyuki Uetake

**Affiliations:** 1Department of Breast Surgery, Tokyo Medical and Dental University, Tokyo 1138519, Japan; oda.srg2@tmd.ac.jp; 2Department of Plastic Surgery, Tokyo Medical and Dental University, Tokyo 1138519, Japan; moriplas@tmd.ac.jp (H.M.); noriplas@tmd.ac.jp (N.U.); 3Department of Gynecology, Tokyo Medical and Dental University, Tokyo 1138519, Japan; k.wakana.crm@tmd.ac.jp (K.W.); noshcrm@tmd.ac.jp (N.O.); 4Department of Gastrointestinal Surgery, Tokyo Medical and Dental University, Tokyo 1138519, Japan; tokunaga.srg1@tmd.ac.jp (M.T.); yusatoh.srg1@tmd.ac.jp (Y.S.); 5Department of Specialized Surgeries, Graduated School, Tokyo Medical and Dental University, Tokyo 1138519, Japan; hayashi.srg2@tmd.ac.jp (K.H.); kumaki.srg2@tmd.ac.jp (Y.K.); ishi.srg2@tmd.ac.jp (T.I.); okasrg2@tmd.ac.jp (K.O.); h-uetake.srg2@tmd.ac.jp (H.U.)

**Keywords:** breast cancer, laparoscopic abdominal surgery, primary breast reconstruction, abdominal flap

## Abstract

*Background and objectives*: Our department has been performing primary breast reconstruction for breast cancer surgery, incorporating a transverse rectus abdominis myocutaneous flap (TRAM)/vertical rectus abdominis myocutaneous flap (VRAM) since 1998 and a deep inferior epigastric artery perforator flap (DIEP) since 2008. Currently, most gastrointestinal operations in abdominal surgery are performed laparoscopically or are robot-assisted. Cases in which abdominal surgery was performed after breast reconstruction using an abdominal flap were reviewed. *Method:* A total of 119 cases of primary breast reconstruction using an abdominal flap performed in our department were reviewed. *Result:* The reconstructive techniques were DIEP in 69 cases and TRAM/VRAM in 50 cases. After breast surgery, seven abdominal operations were performed in six cases. In DIEP cases, one robotic surgery was performed for uterine cancer, and one laparoscopic surgery was performed for ovarian tumor. In TRAM/VRAM cases, two laparoscopic cholecystectomies, one laparoscopic total gastrectomy, one laparoscopic ileus reduction, and one open total hysterectomy oophorectomy were performed. Six surgeries were completed by laparoscopy or robotic assistance. *Conclusion:* The survival rate after breast cancer surgery is improving, and the choice of breast reconstruction procedure should take into account the possibility of performing a prophylactic resection of the ovaries due to the genetic background and possibly postoperative abdominal surgery due to other diseases. However, in cases in which laparoscopic surgery was attempted after breast reconstruction using an abdominal flap, the laparoscopic surgery could be completed in all cases.

## 1. Introduction

Since the reports of TRAM in 1982 and DIEP in 1994, abdominal autologous breast reconstruction surgery for breast cancer has become a common procedure [1,2]. In Japan, however, the surgery was somewhat slower to spread. The number of breast cancer cases in Japan increased in the 2000s, and it is now the most common cancer in Japanese women [3].

Our department has been performing primary breast reconstruction for breast cancer surgery since 1998 with TRAM/VRAM reconstruction and since 2008 with DIEP reconstruction. On the other hand, in abdominal surgery, cholecystectomy was mainly performed laparoscopically in the 1990s, but today most gastrointestinal operations are performed laparoscopically or are robot-assisted.

The survival rate has increased because of improved breast cancer treatment [3]. Therefore, abdominal surgery may be required after breast reconstruction using abdominal tissue. Since there have been few previous reports of patients who have undergone primary breast reconstruction with abdominal autologous tissue and who then underwent laparoscopic or robot-assisted abdominal surgery, the experience of our institution is reported.

## 2. Methods

A total of 119 patients who were diagnosed with breast cancer at our institution from 1998 to 2018 underwent primary breast reconstruction by TRAM/VRAM or DIEP. The median follow-up was 9.7 years (1.6–21 years). Patients’ background characteristics are shown in Table 1; overall survival curves for the 119 cases are shown in Figure 1. Six of these patients who underwent abdominal surgery after breast cancer surgery were reviewed.

Survival curves were drawn by the Kaplan-Meier method. EZR (Saitama Medical Center, Jichi Medical University, Saitama, Japan), which is a graphical user interface for R (The R Foundation for Statistical Computing, Vienna, Austria), was used for this analysis [4].

Our policy on port placement for laparoscopic or robotic surgery after abdominoplasty is as follows, and Figure 2a shows the schema for TRAM/VRAM reconstruction. The rectus abdominis myocutaneous flap is folded back at the cardiac fossa, so placing a port here should be avoided. In addition, prolonged pressure during laparotomy should be avoided in this area. The rectus abdominis defect is just a suture of the anterior sheath of the rectus abdominis muscle, so that a port should not be placed there. Inserting the port by splitting the umbilicus vertically should be avoided if possible, but if it could be split vertically in two, necrosis would not occur.

Figure 2b shows the schema after DIEP reconstruction. Although it does not require as much attention as the rectus abdominis myocutaneous flap, the anterior sheath of the rectus abdominis muscle was incised and sutured to harvest the graft for the DIEP flap. The port should not be inserted into this suture. The handling of the umbilicus is the same as after TRAM/VRAM reconstruction.

## 3. Results

Abdominal surgery was performed seven times in six cases. Of these, six operations in five cases were performed laparoscopically or robot-assisted. The median BMI at the six laparoscopic surgeries in five cases was 27.1 (range: 20.6–30.9) kg/m^2^.

Robot-assisted surgery was performed for uterine cancer in one case. Laparoscopic surgery was performed for an ovarian cyst in one case, gastric cancer in one case, cholelithiasis in two cases, and ileus reduction in one case. One open laparotomy was performed for ovarian cancer, which was accompanied by an abdominal incisional hernia. These six cases are shown in Table 2. There were no cases of prophylactic ovariectomy.

For surgery on the pelvic organs, uterus, and ovaries, an insufficient extension of the lower abdomen was not much of a problem. The surgical field of view was good because the operation was mostly completed in the pelvis (Figure 3). The resected uterus and ovaries were removed from the body through the vagina. In gynecologic surgery, ports are usually not inserted in the epigastric area. The port should be placed avoiding the umbilicus.

The upper abdominal operations, total gastrectomy, and cholecystectomy were relatively adequate in terms of working space. Particularly for gastric cancer surgery, adequate lymph node dissection and intestinal anastomosis in the abdominal cavity were possible (Figure 4a,b). Although port penetration may be difficult in ileus surgery, depending on the degree of bowel dilatation, the intra-abdominal space was accessible, and the field of view was good in the present case (Figure 4c,d). The adherent band was resected, and the ileus could be released.

When performing laparoscopic cholecystectomy, the port was inserted with careful attention to the folded portion of the TRAM in the epigastric area. The upper abdominal field of view was good (Figure 5).

## 4. Discussion

In Japan, abdominoplasty is mainly performed for breast reconstruction, whereas in Europe and the United States, it is performed for the correction of rectus diastasis and obesity, as well as breast reconstruction [5,6]. Especially when performing bariatric surgery after abdominoplasty, e.g., laparoscopic sleeve gastrectomy, there are difficulties with pneumoperitoneum and the lack of working space, and the number of ports may be increased to compensate for this [7,8].

There are three problems in laparoscopic abdominal surgery after breast reconstruction with an abdominal flap. They are the availability of working space in the abdominal cavity, the position of port insertion, and the risk of abdominal incisional hernia. With regard to upper abdominal and pelvic surgery after breast reconstruction with abdominal autologous tissue, the procedure could be performed with little or no problems. In all cases, the pneumoperitoneum pressure was performed as normal. There were no cases of colorectal surgery in the present study. Atallah et al. reported 11 cases of laparoscopic colon surgery after abdominoplasty [9]. The report did not state the reason for the abdominoplasty, but all 11 cases were completed laparoscopically [9]. Colorectal surgery requires mesenteric expansion and requires a larger working space than surgery on other internal organs. We would like to examine this in detail if colorectal surgery is performed at our institute in the future. There is an interesting report on laparoscopic surgery after breast reconstruction with an abdominal flap. When Barukin et al. attempted to perform a laparoscopic adrenalectomy in a patient 3 months after DIEP, they gave up due to difficulty with pneumoperitoneum and the lack of working space [7]. However, when they tried again 6 months after DIEP, they succeeded in their second attempt at laparoscopic adrenalectomy and were able to perform a complete resection [7]. Abdominal extension is considered to improve over time. In the present study, it was possible to perform laparoscopic surgery with relatively little difficulty, which may be due to the fact that Japanese people are less obese than Westerners and usually have less intraabdominal fat. The median BMI at six laparoscopic surgeries in five cases was 27.1 (range: 20.6–30.9) kg/m^2^.

Attention should be paid to the port insertion site, but damage to the rectus abdominis fold, especially in cases of VRAM/TRAM, should be avoided. During upper abdominal open surgery, hooking the rectus abdominis flap fold in the epigastric region for a long time should also be avoided. This is because it can cause damage and compression of the blood vessels, which can reduce blood flow to the rectus abdominis flap. Essentially, it is best to avoid making an incision in the umbilicus and inserting a port. However, in all four cases (gastric cancer, ileus, uterine cancer, and ovarian cyst) in which laparoscopic surgery was performed, the port was inserted through the umbilicus in two longitudinal sections, and there was no postoperative obstruction to blood flow in the umbilicus. Therefore, if the umbilicus can be accurately divided into two vertically, postoperative damage may be less likely to occur. There was no single-port, single-hole procedure among the present cases. It is possible that a large single port could cause blood flow obstruction in the divided umbilicus.

Postoperative abdominal incisional hernias have decreased as DIEP has become more common, and the frequency of reconstruction with TRAM/VRAM has decreased. However, in cases of complicated abdominal incisional hernias, open surgery may be better. Even small hernias are likely to be aggravated by pneumoperitoneum. At the same time, it is possible to repair an abdominal incisional hernia.

Based on these findings, the following points should be kept in mind when placing a port for laparoscopic surgery after an abdominal flap. First, after TRAM/VRAM, prolonged pressure should not be applied to the origin of the rectus abdominis muscle in the cardiac fossa (the fold of the rectus abdominis muscle). In addition, a port should not be placed in the rectus abdominis fold. The defect in the rectus abdominis muscle used for the TRAM/VRAM is sutured with the anterior sheath of the rectus abdominis muscle, so port placement should be avoided around this area. Vertical incisions in the umbilicus should be avoided if possible. After DIEP, the anterior sheath of the rectus abdominis muscle is incised and sutured for graft harvesting of the DIEP flap. Port insertion in this area should be avoided if possible.

The survival rate after treatment for breast cancer is increasing, and the choice of a breast reconstruction technique should take into account the possibility of a prophylactic ovarian resection due to the genetic background and, in some cases, postoperative abdominal surgery due to other diseases. In the present cases, laparoscopic or robot-assisted surgery after primary breast reconstruction using an abdominal flap was completed without converting to open surgery.

This report describes only seven abdominal surgeries, and a statistical analysis could not be performed. If the collection of cases from multiple institutions is possible in the future, it may be possible to analyze the results in terms of abdominal diseases or surgical techniques. We hope that this will be an opportunity to increase attention to abdominal surgery after abdominoplasty. Abdominal surgeons have limited experience with breast reconstruction with an abdominal flap. Therefore, several considerations for laparoscopic surgery were described. Information sharing between abdominal surgeons and breast or plastic surgeons about previous abdominoplasty may also be important.

## 5. Conclusions

The choice of breast reconstruction procedure should take into account the possibly postoperative abdominal surgery due to other diseases. In cases in which laparoscopic surgery was attempted after breast reconstruction using an abdominal flap, the laparoscopic surgery could be completed in all cases.

## Figures and Tables

**Figure 1 medicina-57-00952-f001:**
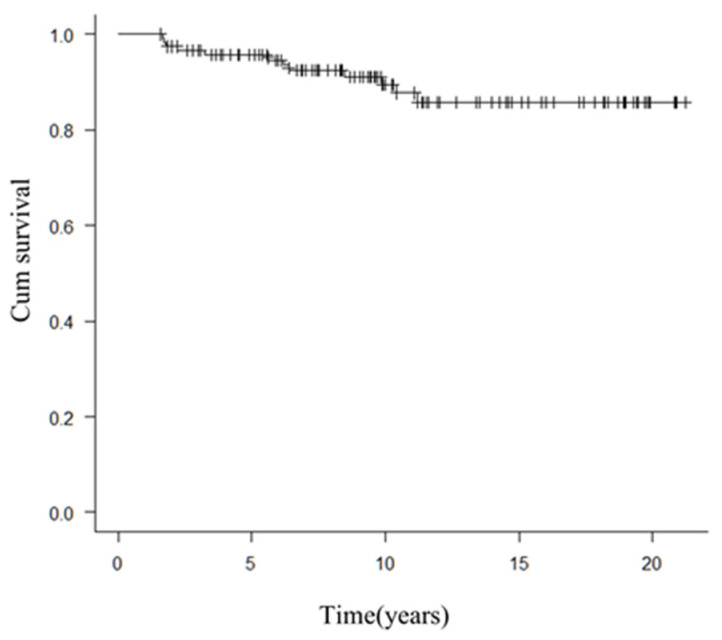
Overall survival of the 119 cases.

**Figure 2 medicina-57-00952-f002:**
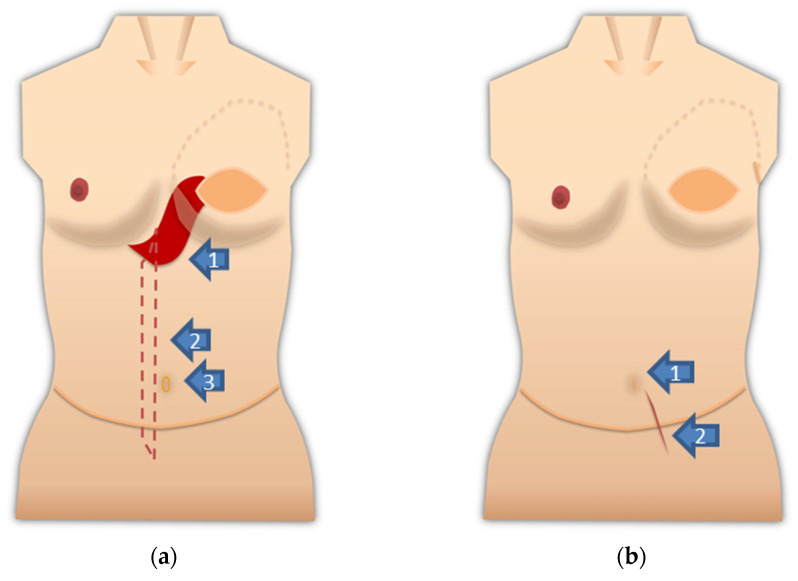
(**a**) Notes on port placement are shown on the schema after VRAM/TRAM reconstruction. Do not apply prolonged pressure to the rectus abdominis fold and avoid port placement (arrow 1: the red area is the fold of the rectus abdominis muscle). There is a defect in the rectus abdominis muscle (arrow 2: red dotted line area). The anterior sheath of the rectus abdominis muscle is sutured. The port should not be placed here. Arrow 3 points to the umbilicus. The umbilicus is split lengthwise into exactly two pieces. Alternatively, a port can be placed transabdominally from the side where the rectus abdominis muscle remains; (**b**) notes on port placement are shown on the schema after DIEP reconstruction. Port insertion in this area should be avoided if possible. The umbilicus should be divided into exactly two parts, or if possible, the port should be placed in a position that will not damage the umbilical root (Arrow 1). The anterior sheath of the rectus abdominis muscle is incised and sutured for graft harvesting of the DIEP flap (arrow 2).

**Figure 3 medicina-57-00952-f003:**
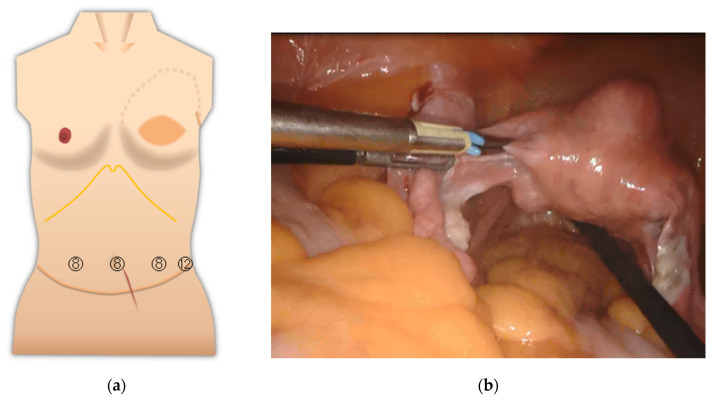
(**a**) Port layout of Case 1. The circled numbers indicate the port diameter (mm, the same applies to the following Figures). The blue line is an incision line; (**b**) view of the pelvic region.

**Figure 4 medicina-57-00952-f004:**
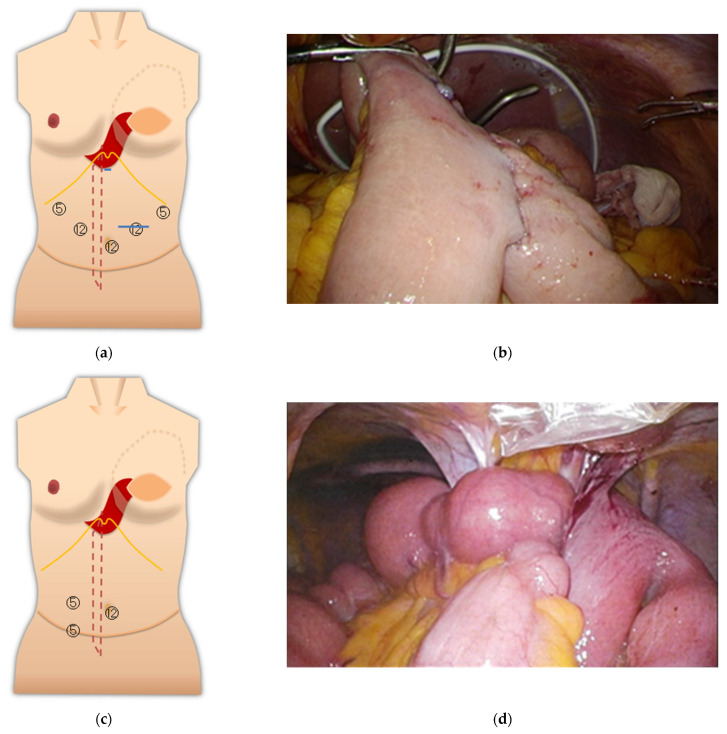
(**a**) Port layout of Case 4 during gastrectomy. The blue line is an incision line; (**b**) view of the abdominal cavity after intestinal anastomosis; (**c**) port layout during ileus reduction surgery; (**d**) working space was adequate.

**Figure 5 medicina-57-00952-f005:**
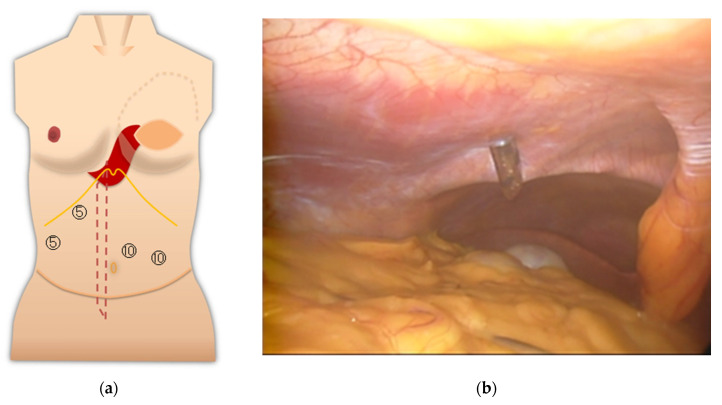
(**a**) Port layout of Case 5; (**b**) inside the abdominal cavity.

**Table 1 medicina-57-00952-t001:** Patients’ background characteristics.

Variable	Number
Breast cancer stage	
0	17
I	55
II	43
III	4
Reconstruction method	
DIEP	69
VRAM/TRAM	50

**Table 2 medicina-57-00952-t002:** Seven cases of abdominal surgery after VRAM/TRAM or DIEP reconstruction.

Case	Breast Surgery			Abdominal Surgery						
	Age	Reconstruction Method	Breast Cancer Stage	Age	Abdominal Disease	Operation Method	Approach	Operation Time	BMI	Figure
1	46	DIEP	I	49	Uterine Body Cancer	Hysterectomy	Robotic	2:51	30.9	3
2	44	DIEP	IIA	45	Ovarian cyst	Ooohorectomy	Laparoscopic	2:57	27.2	
3	46	TRAM	IIA	62	Cholelithiasis	Cholecystectomy	Laparoscopic	0:45	20.6	
4	45	TRAM	IIB	60	Gastric Cancer	Total Gastrectomy	Laparoscopic	7:44	26.9	4
				63	Post-operative Ileus	Resection of an adherent band	Laparoscopic	0:58	20.9	
5	50	TRAM	I	54	Cholelithiasis	Cholecystectomy	Laparoscopic	1:15	29.3	5
6	46	VRAM	I	61	Ovarian cancer	Hysterectomy	Laparotomy	11:01	25.5	
						Ooohorectomy				
						Abdominal Wall Scar Hernia Repair				

TRAM: transverse rectus abdominis myocutaneous flap; VRAM: vertical rectus abdominis myocutaneous flap; DIEP: a deep inferior epigastric artery perforator flap.

## Data Availability

The datasets analyzed in the present study are available from the corresponding author on reasonable request.

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
