# Peer review of "Laparoscopic Abdominal Surgery after Primary Breast Reconstruction Using an Abdominal Flap"

_medicina, 2021, doi:10.3390/medicina57090952_

Round 1

Reviewer 1 Report

The paper is informative, clear, and well written with no unnecessary information.
I have no particular doubts or concerns

Author Response

Thank you for your review and comments. We had some doubts about the safety of abdominal surgery after abdominoplasty, but were able to complete laparoscopic surgery at our own institution. Thank you for the opportunity to report our experience.

Reviewer 2 Report

All in all not very excitung content.

The authors present their cses of breast reconstruction 

and followed up patients how had abdominal surgery later on 

I can not find the major problem of abdominal surgery after

a DIEP and I can not find the evidence for the authors

recommendations of port placement (experience of 7 cases is not much)

Author Response

Thank you for the peer review.

As you pointed out, we have a small number of cases and cannot provide statistical evidence.

As in the previous report, we reported that we were able to complete the laparoscopic surgery after abdominoplasty. Since there has been little mention of the considerations for port placement after abdominoplasty, we have given our opinion here.